# Incidental Indeterminate Renal Lesions: Distinguishing Non-Enhancing from Potential Enhancing Renal Lesions Using Iodine Quantification on Portal Venous Dual-Layer Spectral CT

**DOI:** 10.3390/jpm13111546

**Published:** 2023-10-28

**Authors:** Simone van der Star, Pim A. de Jong, Madeleine Kok

**Affiliations:** Department of Radiology, University Medical Center Utrecht, P.O. Box 85500, 3584 CX Utrecht, The Netherlands; p.dejong-8@umcutrecht.nl (P.A.d.J.); m.kok-16@umcutrecht.nl (M.K.)

**Keywords:** computed tomography, kidney neoplasms, iodine, sensitivity, specificity

## Abstract

The purpose of our study is to determine a threshold for iodine quantification to distinguish definitely non-enhancing benign renal lesions from potential enhancing masses on portal venous dual-layer spectral computed tomography (CT) to reduce the need for additional multiphase CT. In this single-center retrospective study, patients (≥18 years) scanned between April 2021 and January 2023 following the local renal CT protocol were included. Exclusion criteria were patients without renal lesions, lesions smaller than 10 mm, only fat-containing lesions, abscesses or infarction, follow-up after radiofrequent ablation, wrong scan protocol, or artefacts. Scans were performed on a dual layer detector-based spectral CT (CT 7500, Philips Healthcare, Best, The Netherlands). Iodine concentration (mgI/mL) in renal lesions was determined using spectral data. Analyses were performed for all lesions and for lesions of >30 HU on portal venous CT. Enhancement on multiphase CT (≥20 ΔHU from true unenhanced (TUE) to portal venous phase (PVP) CT) was used as reference standard. To determine thresholds for iodine concentration receiver operating characteristic (ROC) curves, area under the curve (AUC) and 95% confidence intervals were calculated. To obtain thresholds for definite (non-)enhancement, 100% sensitivity with maximum specificity and 100% specificity with maximum sensitivity were noted. Data were measured using one reader. To assess interobserver agreement, a second reader performed measurements on the PVP CT scans. A total of 103 patients (62 years ± 14, 68 men) were included. We measured 328 renal lesions, 56 enhancing lesions (17%) in 38 patients and 272 non-enhancing lesions (83%) in 86 patients. The threshold for non-enhancing lesions was 0.76 mgI/mL or lower (100% sensitivity, 76% specificity). The threshold for a definite enhancing mass was 1.69 mgI/mL or higher (100% specificity, 78% sensitivity). A total of 77% of indeterminate lesions (>30 HU on PVP CT) in our study could be definitely characterized. Renal lesions can be definitively classified as non-enhancing or enhancing on PVP spectral CT using thresholds of 0.76 mgI/mL or 1.69 mgI/mL, respectively, eliminating the need for multiphase imaging.

## 1. Introduction

Due to increased use of multidetector computed tomography in the last decades, the incidence of renal incidentalomas increased as almost half of people of >50 years had a renal lesion [1]. Nowadays, more than 50% of malignant renal masses are diagnosed incidentally [2]. They are frequently encountered in single-phase acquired abdominal computed tomography (CT) scans performed for other reasons. However, to assess (non-)enhancement and differentiate between benign lesions and possible malignant masses, multiphase CT is necessary.

According to the Bosniak classification, which was updated in 2019, homogeneous lesions of −9 to 20 HU or ≥70 HU in non-contrast CT and lesions of ≤30 HU in portal venous phase (PVP) CT can be classified as benign cysts [3]. However, for lesions of >30 HU at PVP CT it is not possible to differentiate between a hyperattenuating cyst or an enhancing mass. These indeterminate lesions require further diagnostics, e.g., follow-up, further multiphase CT, magnetic resonance imaging, or histological proof through biopsy or surgery.

Dual-energy techniques could reduce the need for further (non-)invasive diagnostics as it allows quantification of iodine on single-phase, post-contrast CT. This is possible due to use of different energy spectra as each material has a specific energy-dependent attenuation curve. When using single energy CT, different materials (i.e., calcium and iodine) may show similar attenuation values. However, on an additional (lower) energy spectrum these materials will have different attenuation values. This concept allows differentiation and quantification of materials; for further explanation we refer to a review of McCollough et al. [4].

Using postprocessing algorithms, iodine can be selectively displayed or eliminated, leading to iodine-only and virtual non-contrast series (VNC), respectively. To quantify enhancement of a renal lesion using spectral CT, both iodine-only and VNC series can be used. In the literature there is contradictory evidence about the agreement of measured HU-values on VNC series and true unenhanced (TUE) CT [5,6,7,8]. But, a recently published paper found a strong correlation between VNC and TUE [9]. However, the potential enhancement is not taken into account using this method. The other approach is to quantify iodine uptake in a lesion on iodine-only series, where enhancement can be defined as a threshold expressed as milligram iodine per milliliter (mgI/mL).

Several previous studies established optimal thresholds of iodine concentration to detect enhancement, ranging from 0.5 to 1.3 mgI/mL for dual-source dual-energy CT [10,11,12,13,14] and from 1.28 to 2.0 mgI/mL for fast-switching dual-energy CT [10,11,15]. However, sensitivity and specificity values were not as high as needed in these studies to prevent missing enhancing and, therefore, potentially malignant renal masses.

The purpose of our study is to determine a threshold for iodine quantification to distinguish definitely non-enhancing benign lesions from potential enhancing renal masses on portal venous dual-layer spectral CT to reduce the need for additional multiphase CT.

## 2. Materials and Methods

### 2.1. Ethic Statement

A waiver of written informed consent was obtained due to the retrospective study design (Vidatum reference number 23U-0217).

### 2.2. Study Population

In this retrospective study, all patients with an age of 18 years or older that were scanned for clinical indication according to the local renal tumor protocol between April 2021 and January 2023 were included consecutively. Exclusion criteria were patients without renal lesions, lesions smaller than 10 mm due to possible partial volume artefacts and pseudoenhancement [16,17], only fat-containing lesions (≤−20 HU on TUE CT or known angiomyolipomas [18]), abscesses or infarction, follow-up after radiofrequent ablation, wrong scan protocol, or artefacts. In the category of artefacts, we also included patients that positioned their arms alongside the body as this causes beam hardening artefacts making attenuation measurements unreliable, especially for smaller lesions.

If patients had multiple renal lesions, a maximum of 10 lesions were included, starting with selecting the enhancing lesions. The study flowchart is shown in Figure 1.

### 2.3. Image Acquisition and Reconstruction

Scans were performed on a dual-layer detector-based spectral CT (CT 7500, Philips Healthcare, Best, The Netherlands). Patients were scanned according to the local renal tumor protocol. This protocol consists of an unenhanced CT, after which iodinated contrast is administered (single phase injection) and an arterial, portal venous, and (optional) equilibrium phase is acquired. Scan timing was performed using bolus tracking. A ROI was placed in the proximal abdominal aorta with a threshold of 150 HU. The threshold-delay was 20 s for the arterial phase and 90 s for the PVP. For the purpose of this study only TUE and PVP CT were used, as true enhancement is usually defined on PVP when compared to TUE. The scan range for TUE CT was set from 1 cm cranial of the diaphragm to just caudal of the kidneys. In PVP, the scan range was extended down to the symphysis pubis.

Acquisition parameters are listed in Table 1 and were as follows: collimation of 64 × 0.625 mm, gantry rotation time of 0.27 s, and a pitch of 1.15 and 1.258 for TUE and PVP CT, respectively. The scans were performed with a tube voltage of 120 kV, quality reference tube currents of 25 mAs for TUE CT and 109 mAs for PVP CT, and a CTDIvol of 1.9 and 10.4 mGy, respectively. Spectral images were reconstructed into a slice thickness of 1 mm with increments of 1 mm. All datasets were reconstructed using iterative reconstruction (IMR level 1) and soft (B) kernel. For data analysis, only IMR reconstructions were used. Due to use of detector-based spectral CT scanners, spectral data were generated retrospectively using IntelliSpace Portal version 11 (Philips Healthcare, Best, The Netherlands).

### 2.4. Contrast Material

All patients received iodinated contrast (Ultravist, Iopromide 300 mgI/mL; Bayer Healthcare, Berlin, Germany) via an 18–20 G cannula. Pre-warmed contrast (37 °C) was injected using a CT injection system (MEDRAD Centargo, Bayer Healthcare, Berlin, Germany). Dedicated contrast injection software (Certegra™ P3T; Bayer Healthcare, Berlin, Germany) was used to obtain individualized injection protocols based on body weight and scan duration [19,20]. Based on these parameters, the software adapted contrast volume and flow rate to provide a similar injection duration for each patient. Contrast volume was based on body weight (0.4 g iodine per kg) with a minimum and maximum total dose of 20 and 48 g iodine.

### 2.5. Image Analysis

For our study, both conventional and spectral data were collected. From multiphase CT images, HU values were measured on TUE and PVP CT. To quantify enhancement of a renal mass using spectral CT, two methods are available. The first method is based on difference in HU value between VNC and PVP. A difference of ≥20 HU is defined as enhancing according to the Bosniak classification [3]. The other approach is to quantify iodine uptake in a mass on iodine-only series, where enhancement can be defined as a certain threshold expressed as mgI/mL. Spectral CT images were generated in the IntelliSpace Portal. Iodine density series were used to measure the iodine concentration of renal lesions, expressed as mgI/mL. In dual-layer spectral CT, iodine density is the series in which iodine is selectively displayed.

Measurements were performed by placing circular ROIs covering approximately two-thirds of the renal lesion while avoiding the periphery and/or cortex. In complicated cysts, calcifications and thin septa were avoided. The location and size of the ROIs were kept the same for TUE CT, PVP CT images, and spectral data. Figure 2 shows example images of ROI measurements on conventional and spectral images. The size (maximal diameter in axial plane) and location (cortical, exofytic, or parapelvic) of the lesions were also recorded. An exophytic lesion was classified as such when more than 50% of the lesion was located exophytic, otherwise it was classified as cortical. The measurements were performed by a radiology resident, S.v.d.S., who had three years of CT-experience and was trained and supervised by a board certified radiologist, M.K., who had ten years of CT-experience. To assess interobserver agreement, all PVP scans were measured using a second reader (M.K.) who was blinded to the results of the first reader.

### 2.6. Statistical Analysis

Baseline characteristics of patients and renal lesions were analyzed using descriptive statistics. Continuous variables were described as mean values and categorical variables as frequencies and percentages. Attenuation (HU) on multiphase CT and iodine concentration (mgI/mL) in spectral CT were reported as median values and interquartile ranges as this variable was not normally distributed. Enhancement on multiphase CT was defined as a difference of ≥20 ΔHU between TUE and PVP CT according to the Bosniak classification [3]. Comparative analysis between enhancing and non-enhancing lesions was performed by using the Mann–Whitney U test.

Included renal lesions were categorized as simple cysts (0–19 HU), hyperattenuating cysts (≥70 HU), and potential renal masses (20–69 HU) based on TUE CT [3]. When in clinical practice a potential renal mass is encountered on single phase post-contrast CT, the first step will be measuring attenuation on conventional CT images. According to the Bosniak classification, lesions with an attenuation of ≤30 HU on PVP CT are considered as pseudoenhancement and categorized as Boskniak II. This category is described as highly likely to be benign, requiring no follow-up [3]. As a consequence, in clinical practice only indeterminate lesions of >30 HU at PVP CT have to be analyzed on spectral CT images. Therefore, we performed subanalyses of interdeterminate lesions of >30 HU on PVP CT in addition to analyses of all lesions.

Box and whisker plots were used to depict the relationship between enhancement on multiphase CT and iodine concentration for the categories of simple cysts, hyperattenuating cysts, and potential renal mass (based on TUE CT), and for the relationship between quantity of enhancement on multiphase CT and iodine concentration. To determine thresholds for iodine concentration to identify (non-)enhancing lesions, receiver operating characteristic (ROC) curves, their area under the curve (AUC), and 95% confidence intervals were calculated. We used enhancement on multiphase CT as reference standard.

To determine thresholds for definitely non-enhancing and enhancing lesions, iodine concentration with 100% sensitivity and maximum specificity and 100% specificity and maximum sensitivity were noted, respectively. To calculate the optimal threshold, Youden’s index (sensitivity + specificity − 1) and coordinate closest to the left upper corner in the ROC curve (calculated using the square root of ((1 − sensitivity)^2^ + (1 − specificity)^2^)) were used. The thresholds were determined for all lesions and for indeterminate lesions of >30 HU in PVP CT. Interobserver agreement between measurements on PVP CT scans was calculated using a two-way mixed intraclass correlation coefficient.

Statistical significance was defined as *p* < 0.05. Analyses were performed using IBM SPSS statistical software, version 29.

## 3. Results

### 3.1. Subjects

From a total of 160 patients, 57 patients were excluded due to the following reasons: absence of renal lesions (*n* = 20; most of these patients underwent screening for renal or transitional cell carcinoma and some were referred for suspected renal lesions on ultrasound), lesions smaller than 10 mm (*n* = 6), only fat-containing lesions (*n* = 22), abscess or infarction (*n* = 2), follow-up after radiofrequent ablation (*n* = 3), wrong scan protocol (*n* = 3), and artefacts (*n* = 1). A total of 103 patients with 328 lesions were included, 56 enhancing lesions (17%) in 38 patients and 272 non-enhancing lesions (83%) in 86 patients. Sixty-six patients (64%) had more than one renal lesion. The study flowchart is shown in Figure 1.

Baseline characteristics of patients and lesions are listed in Table 2. The mean age was 62 years (SD: 14; range: 24–86) and 66% were male (*n* = 68). The mean lesion size was 24.9 mm (SD: 17.8). The majority (57.9%) was located cortically and 37.6% of the lesion were exophytic. Based on TUE CT, 54% could be categorized as a simple cyst (0–19 HU), 3.7% as a hyperattenuating cyst (≥70 HU), and 42.3% as a potential renal mass (20–69 HU). On PVP CT, 117 out of 328 lesions (35.7%) could be classified as indeterminate lesions (>30 HU).

The Interobserver agreement between the two readers was excellent (ICC: 0.997; *p* < 0.001).

### 3.2. Relationship between Enhancement on Multiphase CT and Iodine Concentration

The relationship between enhancement on multiphase CT and iodine concentration for simple cysts, hyperattenuating cysts, and potential renal masses (based on TUE CT) is shown in Appendix A. From the category of simple cysts (0–19 HU on TUE CT), three out of one hundred seventy-seven lesions (1.7%) showed enhancement on multiphase CT. One lesion proved to be a lipid-poor angiomyolipoma on follow-up MRI, and one lesion was stable (12 mm) and showed no enhancement on follow-up CT in 24 months. The other lesion (11 mm) showed enhancement of 30 ΔHU on multiphase CT with an iodine concentration of 1.21 mgI/mL. No follow-up imaging has been performed as this lesion was classified by the attending radiologist as a simple cyst based on TUE CT. None of the twelve hyperattenuating cysts showed enhancement. For potential renal masses (20–69 HU on TUE CT) the iodine concentration of enhancing lesions (*n* = 53) and non-enhancing lesions (*n* = 86) was statistically different with a median of 2.82 (Q1–Q3: 1.84–3.62) and 0.39 (Q1–Q3: 0.25–0.53) mgI/mL (*p* < 0.001), respectively.

Appendix A depicts the relationship between the quantity of enhancement on multiphase CT and iodine concentration.

### 3.3. Diagnostic Accuracy of Iodine Quantification

Table 3 gives an overview of the median attenuation on multiphase CT and corresponding median iodine concentration of non-enhancing and enhancing lesions for “all lesions” and “indeterminate lesions of >30 HU at PVP CT”.

Figure 3 shows ROC curves for the diagnostic performance of iodine concentration to differentiate non-enhancing from enhancing lesions. AUC was 0.990 (95% CI: 0.981–0.998) for all lesions (Figure 3a) and 0.974 (95% CI: 0.953–0.996) for lesions of >30 HU at PVP CT (Figure 3b). Sensitivity and specificity for different thresholds of iodine concentration to detect (non-)enhancement are shown in Table 4. The threshold with 100% sensitivity to detect definitely non-enhancing lesions was 0.79 mgI/mL with a specificity of 87% for all lesions and 0.76 mgI/mL with a specificity of 76% for indeterminate lesions of >30 HU at PVP CT. The threshold for 100% specificity to define definitely enhancing lesions was 1.69 mgI/mL for both groups with a sensitivity of 77% and 78%, respectively (Table 4).

When including all lesions, the optimal threshold to detect enhancement was 1.19 mgI/mL (sensitivity 93%; specificity 96%). For lesions of >30 HU at PVP CT, the optimal threshold was slightly higher with 1.33 mgI/mL (sensitivity 91%; specificity 92%). The optimal thresholds were similar for both the Youden’s index and coordinate closest to the left upper corner method.

When applying our thresholds of 0.76 mgl/mL (non-enhancing lesions) and 1.69 mgI/mL (enhancing lesions) in the group of indeterminate lesions (>30 HU at PVP CT; *n* = 117), we can define 47 lesions as non-enhancing (40%), 43 lesions as enhancing (37%), and only 27 lesions as potentially enhancing (23%) on PVP CT. This eliminated the need for additional multiphase CT for 90 lesions (77%) in our patient population with indeterminate lesions (>30 HU on PVP CT). These 90 lesions were found in 50 out of 67 patients (75%).

## 4. Discussion

Our study demonstrated that dual-layer spectral CT is an accurate method to detect (non-)enhancing renal lesions on single-phase, post-contrast CT. Our data suggests that a renal lesion on PVP CT is definitely non-enhancing and benign with an iodine concentration of 0.76 mgI/mL or lower and that a concentration of 1.69 mgI/mL or higher is proof of an enhancing mass. For our indeterminate lesions of >30 HU on PVP CT this could prevent multiphase imaging in 50 out of 67 patients (75%; 90/117 lesions (77%)).

To our knowledge, this the first study providing thresholds for iodine concentration to detect definitely non-enhancing and enhancing renal lesions on portal venous dual-layer spectral computed tomography. This technology can clearly reduce the need for additional multiphase imaging in the future for incidental renal lesions on PVP CT. Several previous studies established optimal thresholds (highest sum of sensitivity and specificity) of iodine concentration to detect enhancement [10,11,12,13,14,15]. However, to prevent missing enhancing and potentially malignant renal masses, thresholds with 100% sensitivity are necessary. Only one prior study (Marin et al. [13]) established a threshold with 100% sensitivity and found a threshold of 1.9 mgI/mL. This is significantly higher than our threshold of 0.76 mgI/mL for definitely non-enhancing lesions. However, they used another dual-energy technique (fast-switching kV) and outcomes of prior studies showed that different dual-energy techniques produce substantially different thresholds [10,11,12,13,14,15]. The optimal thresholds of previous studies for enhancement ranged from 0.5 to 1.3 mgI/mL for dual-source dual-energy CT [10,11,12,13,14] and from 1.28 to 2.0 mgI/mL for fast-switching dual-energy CT [11,12,15], compared to 1.33 mgI/mL for dual-layer technology in our study. Several factors contribute to heterogeneity of these thresholds. First, the chosen optimal threshold is in part subjective and depends on also on study methods such as inclusion criteria. For example, included lesions can involve only hyperattenuating lesions, all lesions, or renal cell carcinomas and complex cysts. Also the definition, of enhancement varies, for example, ≥15 vs. ≥20 ΔHU. Furthermore, available dual-energy systems differ in terms of technical aspects, e.g., spatial and temporal resolution, spectral separation and delay, and postprocessing techniques. At last, each dual energy system and vendor has its own postprocessing software. It is evident that caution is needed with the extrapolation of iodine density thresholds to other vendors and maybe even to different CT generations.

Jacobsen et al. [21] is the only prior study that also investigated thresholds of iodine concentration with dual-layer spectral CT in a phantom. However, they established lower limits of iodine detection and quantification and not thresholds to detect definite (non-)enhancement. They found a limit of detection of 0.55 mgI/mL and limit of quantification of 1.0 mgI/mL, which is fairly different from our patient study. The limit of quantification was defined as “the lowest concentration where the coefficient of variation was below 20%” [21]. It appears that this phantom study cannot easily be translated to patient care.

Although VNC data could also be obtained from our spectral CT data, we decided not to use this method as contrast-enhanced scans were used—providing a straightforward and easy way to acquire iodine density measurements. We consider an approach like VNC, which overlooks possible enhancement, to be less than ideal in this viewpoint. The delta enhancement closely mirrors the iodine density and mandates an unenhanced scan along with the positioning of several ROIs. Our demonstration indicates that by utilizing a single reconstruction and a solitary ROI, the majority of patients’ cases can be solved in the future.

In our study, renal lesions smaller than 10 mm were excluded due to possible partial volume artefacts and pseudoenhancement. Two prior studies also showed that lesions of 15–20 mm are more susceptible for pseudoenhancement [16,17]. In our study only two out of 65 small simple cysts (≤15 mm) based on TUE CT showed enhancement on PVP CT. These lesions measured 11 and 12 mm. The lesion of 12 mm was stable and showed no enhancement on follow-up CT, indicating pseudoenhancement on the initial CT. It is likely that the other lesion of 11 mm also showed pseudoenhancement due to the small size.

Moreover, one lesion that was initially categorized as a simple cyst on TUE CT proved to be a lipid-poor angiomyolipoma on follow-up MRI. Due to microscopic fat, ROI measurements of these lesions will not show negative HU values on TUE CT and may result in low, non-negative HU values on TUE CT. As lipid-poor angiomyolipomas can enhance as vividly as renal cell carcinomas, it is not possible to differentiate with certainty between a lipid-poor angiomyolipoma or renal cell carcinoma using spectral CT [22,23].

Our thresholds slightly differed for “all lesions” and “lesions of >30 HU at PVP CT”. In clinical practice, incidentalomas will be mostly found on PVP CT; therefore, we prefer thresholds from the analysis that excludes renal lesions, which can already be classified as highly likely to be benign on portal venous CT images (≤30 HU at PVP CT) [3].

Our study has some limitations that need to be addressed. Firstly, we investigated only one dual energy technique. Previous studies showed that thresholds to detect enhancement using iodine concentration varied substantially between different dual-energy techniques. As a consequence, our thresholds are only applicable to dual-layer spectral CT scanners, although high accuracy of iodine quantification is also expected for different dual energy methods. Secondly, only a minority (17%) of included lesions are enhancing lesions. However, this ratio of enhancing and non-enhancing renal lesions reflects clinical practice as the majority of incidentalomas are benign. Thirdly, we focused on enhancement on PVP CT and did not analyze enhancement in the arterial phase as in clinical practice most incidentalomas will be found on PVP CT. Moreover, homogeneous lesions of 21 up to 30 HU on PVP CT are classified as benign according to the Bosniak classification [3]. Furthermore, we did not analyze equivocal enhancement (10–20 ΔHU) for which additional MRI may be indicated to differentiate between pseudoenhancement or minimally enhancing lesions like papillary renal cell carcinoma [24]. However, equivocal enhancement is not implemented in the Bosniak classification which is the leading guideline in clinical practice in the Netherlands to stratify the risk of malignancy of (cystic) renal lesions [3]. Another limitation is the exclusion of lesions smaller than 10 mm in our study. We excluded these small lesions as previous studies showed that lesion size smaller than 10 to 15 mm is an independent predictor for pseudoenhancement [16,17], partly due to partial volume artefacts. Lastly, the reference standard of this study was enhancement on multiphase CT and not a reference standard like follow-up or histopathology. However, the presence of enhancement is a major criterion in deciding whether biopsy or surgery of a renal lesion is necessary [25]. So, to reduce additional multiphase imaging for renal incidentalomas our study design fits its purpose.

In conclusion, according to our study, at the thresholds of 0.76 mgl/mL and 1.69 mgI/mL on portal venous dual-layer spectral CT a renal lesion can be determined to be definitely non-enhancing or enhancing, precluding the need for multiphase imaging.

## Figures and Tables

**Figure 1 jpm-13-01546-f001:**
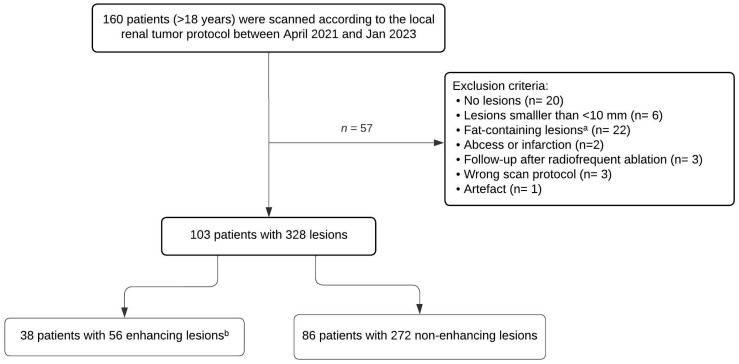
Study flowchart for the inclusion of patients. ^a^: Fat-containing lesions: known angiomyolipoma or ≤−20 HU on true unenhanced CT [18]. ^b^: Enhancing lesion: ≥20 ΔHU between true unenhanced and portal venous phase CT.

**Figure 2 jpm-13-01546-f002:**
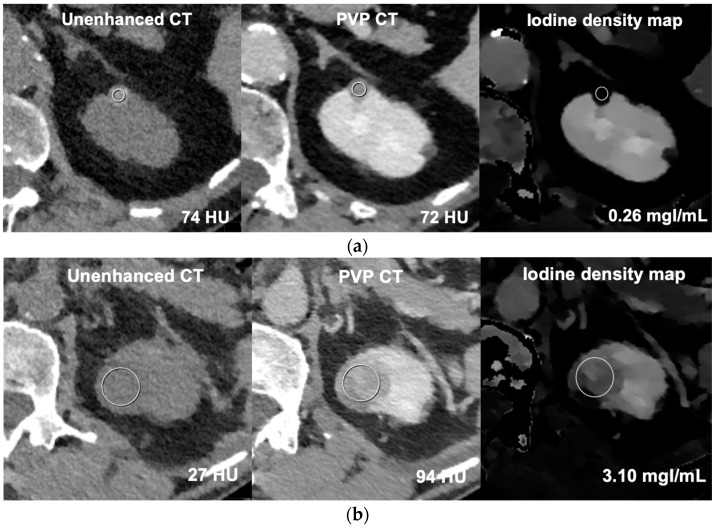
Example CT images of ROI measurements (circles) on conventional CT and spectral CT in an hyperattenuating cyst (**a**) and enhancing mass (**b**). (**a**) Unenhanced CT shows a hyperdense mass in the upper pole of the left kidney of a 69-year-old female without enhancement on portal venous phase CT and a corresponding low iodine concentration on the iodine density map; (**b**) Multiphase CT shows an enhancing mass (67 ΔHU) in the upper pole of the left kidney of a 69-year-old male with a corresponding high iodine concentration on the iodine density map.

**Figure 3 jpm-13-01546-f003:**
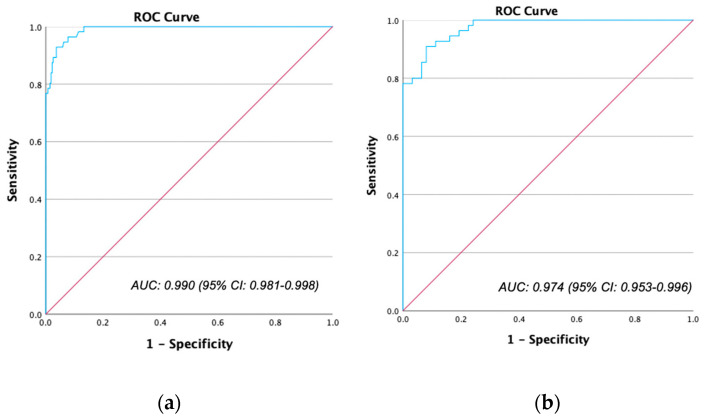
ROC curve for differentiating non-enhancing from enhancing lesions with iodine concentration for all lesions (272 non-enhancing and 56 enhancing) (**a**) and indeterminate lesions of >30 HU (**b**) in portal venous phase CT (62 non-enhancing and 55 enhancing).

**Table 1 jpm-13-01546-t001:** Acquisition parameters of the dual-layer spectral CT scanner ^a^.

	Unenhanced CT	Portal Venous Phase CT
Collimation (mm)	64 × 0.625 mm	64 × 0.625 mm
Tube voltage (kV)	120	120
Tube current (mAs)	Ref: 25 Min/max: 20/none	Ref: 109 Min/max: 20/none
Pitch	1.15	1.258
CTDIvol (mGy)	1.9	10.4
Gantry rotation time (s)	0.27	0.27
Reconstruction kernel	B (IMR level 1)	B (IMR level 1)
Slice thickness/increment (mm)	1/1	1/1

^a^: Spectral CT 7500, Philips Healthcare, Best, The Netherlands. Abbreviations: CTDIvol, computed tomography volume dose index.

**Table 2 jpm-13-01546-t002:** Baseline characteristics of patients and renal lesions.

Patients	*n* = 103		
Age, years, mean (SD; range)	62 (14; 24–86)		
Gender, *n* (%)			
Male	68 (66.0)		
Female	35 (34.0)		
No. of lesions per patient, *n* (%)			
1	37 (35.9)		
2	13 (12.6)		
3	14 (13.6)		
4	7 (6.8)		
5	8 (7.8)		
6–9	7 (6.8)		
≥10	17 (16.5)		
**Renal lesions**	**All**	**Non-enhancing**	**Enhancing**
	*n* = 328	*n* = 272	*n* = 56
Lesion size, mean, mm (SD)	24.9 (17.8)	24.2 (16.6)	28.4 (22.9)
10–15 mm	123 (37.5)	103 (37.9)	20 (35.7)
≥16 mm	205 (62.5)	169 (62.1)	36 (64.3)
Location, *n* (%)			
Cortical	190 (57.9)	149 (54.8)	41 (73.2)
Exofytic	123 (37.6)	109 (40.1)	14 (25.0)
Parapelvic	15 (4.5)	14 (5.1)	1 (1.8)
True unenhanced CT, *n* (%)			
0–19 HU	177 (54.0)	174 (64.0)	3 (5.4)
≥70 HU	12 (3.7)	12 (4.4)	0 (0.0)
20–69 HU	139 (42.3)	86 (31.6)	53 (94.6)
Portal venous phase CT, *n* (%)			
≤30 HU	211 (64.3)	210 (77.2)	1 (1.7)
>30 HU	117 (35.7)	62 (22.8)	55 (98.3)

**Table 3 jpm-13-01546-t003:** Attenuation on multiphase CT and corresponding iodine concentration of non-enhancing and enhancing lesions for all lesions (A) and indeterminate lesions of >30 HU (B) in portal venous phase CT.

	All Lesions		>30 HU at Portal Venous Phase CT	
	*n* = 328		*n* = 117	
	Non-enhancing lesion	Enhancing lesion ^a^	Non-enhancing lesion	Enhancing lesion ^a^
	*n* = 272	*n* = 56	*n* = 62	*n* = 55
ΔHU on multiphase CT, median (Q1–Q3; range)	0 (−3–4; −18–18)	57.5 (34–82; 22–142)	1 (−2–7; −13–18)	59 (34–84; 22–142)
Iodine concentration (mgI/mL), median (Q1–Q3; range)	0.39 (0.25–0.58; 0.00–1.66)	2.55 (1.73–3.57; 0.80–5.95)	0.45 (0.26–0.74; 0.02–1.66)	2.74 (1.78–3.59; 0.80–5.95

a: Enhancing lesion: ≥20 ΔHU between true unenhanced and portal venous phase CT.

**Table 4 jpm-13-01546-t004:** Sensitivity and specificity for different thresholds of iodine concentration to differentiate non-enhancing lesions from enhancing masses for all lesions and indeterminate lesions of >30 HU in portal venous phase CT.

	Threshold Iodine Concentration (mgI/mL)	Sensitivity (%)	Specificity (%)
All lesions	0.79 ^a^ 1.19 ^b^ 1.69 ^c^	100 93 77	87 96 100
>30 HU at portal venous phase CT	0.76 ^a^ 1.33 ^b^ 1.69 ^c^	100 91 78	76 92 100

a: threshold for 100% sensitivity with maximum specificity. b: optimal threshold according to both the Youden’s index and coordinate closest to the left upper corner on the ROC curve calculated using the square root of ((1 − sensitivity)^2^ + (1 − specificity)^2^)). c: threshold for 100% specificity with maximum sensitivity.

## Data Availability

The data presented in this study are available on request from the corresponding author (S.v.d.S). The data are not publicly available due to ongoing unpublished research.

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
