# Peer review of "Incidental Indeterminate Renal Lesions: Distinguishing Non-Enhancing from Potential Enhancing Renal Lesions Using Iodine Quantification on Portal Venous Dual-Layer Spectral CT"

_jpm, 2023, doi:10.3390/jpm13111546_

Round 1

Reviewer 1 Report

Comments and Suggestions for Authors

The authors present a very nice study on iodine quantification and assessment of enhancement for renal lesions. While limited to one type of dual energy CT, this is nicely designed and a great step towards reducing unnecessary imaging.  The chosen figures provide good examples of the issue at hand. 

Comments on the Quality of English Language

Few typos here and there eg. line 157 categorical misspelled and line 260.

Author Response

Dear reviewer,

We are very thankful for your comments. We extensively reviewed our manuscript for typos and corrected them, e.g., line 18, 40, 156 and 269.

Yours sincerely,

Reviewer 2 Report

Comments and Suggestions for Authors

The manuscript is a clear and interesting description of the determination of a threshold in iodine concentration to differentiate benign and malignant structures in the kidney.

As someone unfamiliar with dual-energy CT I do miss a few sentences, perhaps with a reference, on how the Iodine concentration is calculated from the spectral data from the CT. If all prospective readers familiar with the field are not expecting such a description due to it being common knowledge then maybe it can be excluded.

This connects to another major question which the manuscript as is fails to sufficiently discuss. The iodine concentration in different types of tissue should be a biological property, and it makes no sense that it would differ depending on the instrument used for measuring it. I would appreciate a deeper discussion of why the concentration threshold in this study differs from those previously published. If it depends on the difference in instrumentation, then there must be systemic errors present in the quantification of iodine concentration. What could be the cause for these errors, and which type of instrument is more likely to have a correct quantification?

Author Response

Dear reviewer,

We are very thankful for your valuable comments and incorporated the suggestions and feedback in our revised manuscript. Below we listed the comments (bold text) point-by-point, followed by our responses.

  1. As someone unfamiliar with dual-energy CT I do miss a few sentences, perhaps with a reference, on how the Iodine concentration is calculated from the spectral data from the CT. If all prospective readers familiar with the field are not expecting such a description due to it being common knowledge then maybe it can be excluded.

We added a brief explanation in the introduction section about the concepts of dual energy/spectral CT. Further technical details on dual energy CT are discussed in a review of McCollough et al (reference number 4).

The italic text is inserted in the introduction section (line 51-58):

Dual-energy techniques could reduce the need for further (non-)invasive diagnostics as it allows quantification of iodine on single phase post-contrast CT. This is possible due to use of different energy spectra as each material has a specific energy-dependent attenuation curve. When using single energy CT, different materials (i.e., calcium and iodine) may show similar attenuation values. However, on an additional (lower) energy spectrum these materials will have different attenuation values. This concept allows differentiation and quantification of materials, for further explanation we refer to a review of McCollough et al [4].

Using postprocessing algorithms, iodine can be selectively displayed or eliminated, leading to iodine-only and virtual non-contrast series (VNC), respectively.

  1. This connects to another major question which the manuscript as is fails to sufficiently discuss. The iodine concentration in different types of tissue should be a biological property, and it makes no sense that it would differ depending on the instrument used for measuring it. I would appreciate a deeper discussion of why the concentration threshold in this study differs from those previously published. If it depends on the difference in instrumentation, then there must be systemic errors present in the quantification of iodine concentration. What could be the cause for these errors, and which type of instrument is more likely to have a correct quantification?

We agree that we elaborated insufficiently on the differences between optimal iodine thresholds from previous studies. We agree that iodine uptake is a biological property, but there are technological differences between vendors and also design of studies and inclusion criteria play a role. We added an explanation in which we pointed out differences in methodology and technical aspects.

The italic text is inserted in the discussion section (line 299-307):

The optimal thresholds of previous studies for enhancement ranged from 0.5 to 1.3 mgI/ml for dual-source dual-energy CT [11–15] and from 1.28 to 2.0 mgI/ml for fast-switching dual-energy CT [10–12], compared to 1.33 mgI/ml for dual-layer technology in our study. Several factors contribute to heterogeneity of these thresholds. First, the chosen optimal threshold is in part subjective and depends on also on study methods such as inclusion criteria. For example, included lesions can involve only hyperattenuating lesions, all lesions, or renal cell carcinomas and complex cysts. Also the definition of enhancement varies, for example ³15 vs. ³20 DHU. Furthermore, available dual energy systems differ in terms of technical aspects, e.g., spatial and temporal resolution, spectral seperation and delay, and postprocessing technique. At last, each dual energy system and vendor has its own postprocessing software. It is evident that caution is needed with extrapolation of iodine density thresholds to other vendors and maybe even to different CT generations.

Regarding to which dual energy CT system is most accurate:

In literature, there is somewhat contradictory evidence about which dual-energy technology is most accurate. Chen et al [1]  and Jacobsen et al [2] found that fast-switching and dual-source provided the highest accuracy. However, Kim et al [3] and Pelgrim et al [4] found no differences in accuracy between several dual energy techniques.

References

  1. Chen, Y.; Zhong, J.; Wang, L.; Shi, X.; Chang, R.; Fan, J.; Jiang, J.; Xia, Y.; Yan, F.; Yao, W.; et al. Multivendor Comparison of Quantification Accuracy of Iodine Concentration and Attenuation Measurements by Dual-Energy CT: A Phantom Study. American Journal of Roentgenology 2022, 219, 827–840, doi:10.2214/AJR.22.27753.
  2. Jacobsen, M.C.; Schellingerhout, D.; Wood, C.A.; Tamm, E.P.; Godoy, M.C.; Sun, J.; Cody, D.D. Intermanufacturer Comparison of Dual-Energy CT Iodine Quantification and Monochromatic Attenuation: A Phantom Study. Radiology 2018, 287, 224–234, doi:10.1148/radiol.2017170896.
  3. Kim, H.; Goo, J.M.; Kang, C.K.; Chae, K.J.; Park, C.M. Comparison of Iodine Density Measurement among Dual-Energy Computed Tomography Scanners from 3 Vendors. Invest Radiol 2018, 53, 321–327, doi:10.1097/RLI.0000000000000446.
  4. Pelgrim, G.J.; Van Hamersvelt, R.W.; Willemink, M.J.; Schmidt, B.T.; Flohr, T.; Schilham, A.; Milles, J.; Oudkerk, M.; Leiner, T.; Rozemarijn Vliegenthart, & Accuracy of Iodine Quantification Using Dual Energy CT in Latest Generation Dual Source and Dual Layer CT. Eur Radiol 2017, 27, 3904–3912, doi:10.1007/s00330-017-4752-9/.